# Formation of Intermediate Amylose Rice Starch–Lipid Complex Assisted by Ultrasonication

**DOI:** 10.3390/foods11162430

**Published:** 2022-08-12

**Authors:** Paramee Chumsri, Worawan Panpipat, Ling-Zhi Cheong, Manat Chaijan

**Affiliations:** 1Food Technology and Innovation Research Center of Excellence, School of Agricultural Technology and Food Industry, Walailak University, Thasala, Nakhon Si Thammarat 80160, Thailand; 2Zhejiang-Malaysia Joint Research Laboratory for Agricultural Product Processing and Nutrition, College of Food and Pharmaceutical Science, Ningbo University, Ningbo 315211, China

**Keywords:** rice, starch–lipid complex, fatty acid, ultrasonication, resistant starch

## Abstract

Due to the potential reduction in starch availability, as well as the production of the distinct physico-chemical characteristics of starch in order to improve health benefits, the formation of starch–lipid complexes has attracted significant attention for improving the quantity of resistant starch (RS) content in starchy-based foods. The purpose of this research was to apply ultrasonication to produce intermediate amylose rice (*Oryza sativa* L.) cv. Noui Khuea (NK) starch–fatty acid (FA) complexes. The effects of ultrasonically synthesized conditions (ultrasonic time, ultrasonic amplitude, FA chain length) on the complexing index (CI) and in vitro digestibility of the starch–FA complex were highlighted. The optimum conditions were 7.5% butyric acid with 20% amplitude for 30 min, as indicated by a high CI and RS contents. The ultrasonically treated starch–butyric complex had the highest RS content of 80.78% with a V-type XRD pattern and an additional FTIR peak at 1709 cm^−1^. The increase in the water/oil absorption capacity and swelling index were observed in the starch–lipid complex. The pasting viscosity and pasting/melting temperatures were lower than those of native starch, despite the fact that it had a distinct morphological structure with a high proportion of flaky and grooved forms. The complexes were capable of binding bile acid, scavenging the DPPH radical, and stimulating the bifidobacterial proliferation better than native starch, which differed depending on the FA inclusion. Therefore, developing a rice starch–lipid complex can be achieved via ultrasonication.

## 1. Introduction

The desire to improve the quantity of resistant starch (RS) in starchy food components is growing, in order to improve the health benefits [1]. Several methods for producing RS have been used [2]. The formation of starch–lipid complexes has received considerable attention for increasing the RS content in starchy-based foods, owing to the potential reduction in starch availability, as well as the production of the unique physico-chemical characteristics of starch [3]. The fifth variety of resistant starch has been identified as starch–lipid complexes (RS5). Because the complex is fermented by microorganisms in the large intestine rather than digested in the small intestine, RS helps to modulate the postprandial glycemic response, lowering colon inflammation, and the risk of colon cancer [4]. Due to the rising incidence of chronic diseases linked to nutrition, research boosting the creation of starch–lipid complexes has recently received attention. Previous publications have stated that numerous chemical, physical, and enzymatic techniques have been used to speed up the formation of V-type inclusion complexes [5,6,7,8,9]. The capability of amylose to form complexes with lipids is largely attributed to amylose changing from a coil to helix and targeted molecules entering the cores of the amylose helices during complex development [5]. The primary driving forces behind the development of the amylose–lipid complex were hydrogen bonding and hydrophobicity [6]. The formation of the V-type inclusion helical complexes can alter the utilitarian features of starches, such as limiting amylose retrogradation and staling, preventing amylopectin recrystallization, lowering starch expansion force, and improving the starch resistance to enzymatic hydrolysis [10]. Several of the parameters, including the degree of polymerization, polydispersity, degree of branching, amylose chain length, and starch concentration, as well as lipid structure (fatty acid (FA) chain length, degree of saturation, and FA concentration), influence the molecular organization and physico-chemical properties of amylose–FA complexes [4,5,6,7,8,9,10]. Moreover, the method used to embed the lipid into the amylose helical structure may have altered the degree of complex production and functionality of the resulting starch complex, which were rarely reported in comparison to the influence of the starch and lipid characteristics on complex formation [4,5,6,7,8,9,10].

The physical modifications (such as high-pressure homogenization, high hydrostatic pressure, and ultrasonication) were able to decrease the usage of chemicals and the production of waste, making them deemed secure and safe for the environment. The high-pressure homogenization process, according to Meng [11], encouraged the development of the maize starch–palmitic acid complex and increased the complexity of the complexation. The creation of the amylose–lipid complexes and the slowed digestibility can both be accomplished, according to Chen et al. [12], using the high-pressure homogenization procedure. High hydrostatic pressure (HHP) was used to prepare the amylose–lipid complexes, and Guo et al. [13] suggested that non-thermal methods might be used to quickly produce V-type complexes, using HHP.

Recently, ultrasonication has also been shown to assist in the integration of lipids into starch molecules [6], potentially facilitating the development of V-type complexes. The ultrasonic treatment is a low-cost, high-efficiency, and simple method for dissolving swollen starch, releasing linear amylose chains, improving lipid dispersibility in gelatinized starch, and facilitating the complexation process between amylose and target compounds [14,15], all of which have an effect on the physico-chemical and functional aspects of starch [16]. Previous research has demonstrated that the creation of single helix complexes by ultrasonic treatment is greatly aided by the release of additional linear chains from the enlarged starch granules and the improvement of lipid dispersibility in solution [17]. The scission of amylopectin’s branching points was aided by ultrasound’s destruction of the C–O–C bonds in the α-1,6 glycosidic linkage, which increased the number of linear chains [5]. Raza [9] has described using dual-frequency power ultrasound (20/40 kHz) to create complexes between the arrowhead starch and linoleic acid/stearic acid. The complexes had complexing index (CI) values of more than 65%, and after 40 min of treatment, the maximum CI value was discovered at 83.04%. Herein, the effect was investigated, in relation to the in vitro digestibility, of the ultrasonically assisted formation of the intermediate-amylose Thai indigenous rice starch–lipid complexes with varying FA chain lengths. This finding aided in a better understanding of the alternative use of indigenous hard-grain rice starch as an amylose source for preparing more healthy. starchy functional food ingredients using ultrasonication.

## 2. Materials and Methods

### 2.1. Brown Rice Sample and Starch Preparation

The brown rice (*Oryza sativa* L.) cv. *Noui Khuea* (NK) flour, an indigenous underutilized intermediate-amylose (22%) short-grain hard rice widely cultivated in Southern Thailand [18], was used in this study. The brown rice was soaked in water at a ratio of 1:4 (*w*/*v*) for 6 h at 4 °C before being pulverized in a double-disk stone mill. Then, the slurry was placed into a thick fabric bag and squeezed for 10 min using a hydraulic press. The wet-milled flour was then dried at 60 °C in a hot-air oven (DT 20S tray drier, 4500 W; Owner Foods Machinery Co., Ltd., Bangkok, Thailand) to reduce the moisture content to around 13%. The dried sample was ground to a powder, and sieved through a 100-mesh sieve. The rice flour was kept in ethylene-vinyl alcohol copolymer bags at −20 °C until it was needed.

To extract the starch, the method of Sun et al. [19] was used. In a flask, the rice flour was mixed with distilled water at a ratio of 1:7 (*w*/*v*), then NaOH (0.5 g NaOH per 100 g rice, *w*/*w*) was added. The flask was then placed in a water bath at 40 °C for 3 h with a shaker (125 rpm). The pale-yellow emulsion was filtered through a screen with a mesh diameter of 180 μm after incubation. The upper pale-yellow liquid was removed after centrifuging the filtrate for 20 min at 15,600× *g* (RC-5B plus centrifuge; Sorvall, Norwalk, CT, USA). The precipitation layer was centrifuged for 15 min at the same speed after being rinsed twice with distilled water. After discarding the black section of the upper layer, the precipitate was dispersed in distilled water and centrifuged for another 15 min at 15,600× *g*. To obtain the rice starch, the precipitated layer was moved to an aluminum box and dried at 50 °C for 12 h.

### 2.2. Production of the NK Starch–FA Complex Assisted by Ultrasonication

The first trial was carried out to study the effect of ultrasonic time (0–60 min) on the starch–butyric acid complex at the same amplitude of 20%. The FA (1.5% dissolved in absolute ethanol) (40 mL) was added to the native NK starch dispersion (6 g native starch dispersed in 60 mL distilled water) [6]. The mixture was continuously stirred and heated (90 °C/30 min). The paste was cooled down to 60 °C and then ultrasonically treated at a power of 750 W, a single frequency of 20 kHz, for 0–60 min with amplitude of 20% (Model VCX600; Sonics & Materials Inc., Newtown, CT, USA) and interval pulses of 4 s at room temperature (26–28 °C). The mixture was cooled to 25 °C and centrifuged with an RC-5B plus centrifuge (Sorvall, Norwalk, CT, USA) at 4000× *g* for 10 min. The sample was washed thrice with a 50% ethanol/water mixture, and centrifuged at 4000× *g* for 10 min. The precipitate was dried at 40 °C for 24 h (FD 115; BINDER GmbH, Tuttlingen, Germany). The starch paste without ultrasonication was used as the control. Then, the optimum ultrasonic time was selected for the second trial on the effect of ultrasonic amplitude (0–40%) on the starch–butyric acid complex. Thereafter, the effect of the butyric acid concentration (1–10%) on the starch–butyric acid complex produced under optimum ultrasonic time and amplitude was studied. Finally, the effect of the FA chain length, including butyric acid (C4:0), lauric acid (C12:0), stearic acid (C18:0), and linoleic acid (C18:2) at the optimum concentration on the complexed formation, was investigated using the optimum ultrasonic time and amplitude. The complexing index (CI) [20] and the in vitro digestibility (i.e., rapidly digestible starch (RDS), slowly digestible starch (SDS), and RS content) [21] of all of the samples were determined.

CI represents the percentage of complexation between the starches and lipids [20]. The starch paste (5.0 g) was mixed with 25 mL of distilled water at 50 °C in a 50 mL capped tube. After being vortexed for 2 min, 100 μl of the resulting dispersion was mixed with 15 mL of distilled water, followed by the addition of 2 mL of iodine solution (2.0% KI and 1.3% of I_2_ in distilled water). Then, the absorbance was read at 690 nm. The pastes made only of starch were used as a reference. To avoid starch retrogradation, the tests were conducted within 60 min. The CI was calculated as follows:(1)$$\mathrm{CI}\;=[\;({\mathrm{Abs}}_{\mathrm{Reference}}-{\mathrm{Abs}}_{\mathrm{Starch}-\mathrm{Lipid}})/{\mathrm{Abs}}_{\mathrm{Reference}}]\times 100$$

### 2.3. Characterization of the NK Starch–FA Complex Assisted by Ultrasonication

The physico-chemical and in vitro bioactivity of the starch–lipid complexes with varying FA chain lengths were also investigated. The water absorption capacity (WAC), oil absorption capacity (OAC), swelling and solubility indices, and light transmittance (LT) were determined, according to Ashwar et al. [22].

For the WAC and solubility index, 1 g of flour was suspended in 10 mL of distilled water and mixed with a vortex mixer for 1 min. The suspensions were heated in a water bath at 30 °C for 30 min with gentle stirring and then centrifuged at 1500× *g* for 10 min (RC-5B plus centrifuge). The supernatant was carefully poured into an aluminum moisture can before being dried at 105 °C overnight. The sediments were collected and weighed. The WAC and solubility were calculated using the following formulas:(2)$$\mathrm{WAC}\;(g/g)=\frac{\mathrm{Weight}\;\mathrm{of}\;\mathrm{wet}\;\mathrm{sediment}\;\left(g\right)}{\mathrm{Dry}\;\mathrm{weight}\;\mathrm{of}\;\mathrm{flour}\;\left(g\right)}$$
(3)$$\mathrm{Solubility}\;\left(g/g\right)=\frac{\mathrm{Weight}\;\mathrm{of}\;\mathrm{dried}\;\mathrm{supernatant}\;\left(g\right)}{\mathrm{Dry}\;\mathrm{weight}\;\mathrm{of}\;\mathrm{flour}\;\left(g\right)}\;$$

For the OAC, 1 g of flour was mixed with 10 mL of soybean oil for 1 min. After standing at room temperature (30 min), the centrifugation was applied at 1500× *g* for 10 min (RC-5B plus centrifuge). Thereafter, the surplus oil was decanted while the residue (weight of oil absorbed) was weighed. The calculation of OAC was as follows:(4)$$\mathrm{OAC}\;\left(g/g\right)=\frac{\mathrm{Weight}\;\mathrm{of}\;\mathrm{oil}\;\mathrm{absorbed}\;\left(g\right)}{\mathrm{Weight}\;\mathrm{of}\;\mathrm{sample}\;\left(g\right)}$$

For the swelling index, 1 g of flour was mixed with distilled water (30 mL). After heating (85 °C/30 min) in a W350 Memmert water bath (Schwabach, Germany), the sample was cooled to room temperature and centrifuged at 1500× *g* for 20 min (RC-5B plus centrifuge). The swelling index was estimated as the weight of the paste per weight of the dry sample:(5)$$\mathrm{Swelling}\;\mathrm{index}\;\left(g/g\right)=\frac{\mathrm{Weight}\;\mathrm{of}\;\mathrm{the}\;\mathrm{paste}\;\left(g\right)}{\mathrm{Weight}\;\mathrm{of}\;\mathrm{dry}\;\mathrm{sample}\;\left(g\right)}\;$$

For LT, an aqueous starch suspension (1%) was generated by heating at 90 °C in a W350 Memmert water bath (Schwabach, Germany) for 30 min, while stirring continuously at 75 rpm. The suspension was cooled for 1 h at 30 °C. The samples were kept in a refrigerator for 120 h at 4 °C, and a UV spectrophotometer (Shimadzu, MD, USA) was used to measure transmittance every 24 h at 640 nm against a water blank.

The pasting behaviors were determined by the Rapid Visco Analyser (RVA 4500, Perten Instruments, Stockholm, Sweden). The pasting temperature, peak viscosity, breakdown, setback, and final viscosity were reported as the pasting parameters. Differential scanning calorimeter (DSC) was used to determine the thermal behavior of the starch samples, using a calorimeter (DSC6000; Perkin Elmer, Waltham, MA, USA). The onset temperature (T_o_), peak temperature (T_p_), end temperature (T_end_), and enthalpy of gelatinization (ΔH) were recorded. The X-ray diffraction (XRD) patterns of the samples were analyzed by an X-ray diffractometer (Rigaku Oxford Diffraction, Manchester, UK) [23]. Fourier-Transform infrared (FTIR) analysis was completed, using a Bruker Model Vector 33 FTIR spectrometer (Bruker Co., Ettlingen, Germany). For the scanning electron microscopy (SEM), the samples were imaged with a scanning electron microscope (Gemini SEM, Carl Ziess Microscopy, Germany) at a 5 kV accelerator potential, after being vertically coated with gold-palladium.

For the in vitro bioactivities, the bile acid-binding capacity of the starch sample was determined with the colorimetric method described by Ashwar et al. [22]. A 200 mL solution of cholic acid was made by mixing 4.7 mL of 0.1 N NaOH and 200 mg of cholic acid with distilled water. A test tube containing 25 mg of starch and 10 mL of cholic acid solution was then filled. The mixture was agitated for 2 h at 37 °C before being filtered using a 0.2 μm syringe filter. The resultant solution (1 mL) was treated with 1 mL of 0.9% alcoholic furfural solution and 5 mL 16 N sulfuric acid, before being held in an ice bath for 5 min, a 70 °C bath for 8 min, and then another 2 min in an ice bath. Thereafter, the absorbance was read at 490 nm.

The 1,1-diphenyl-2-picrylhydrazyl radical (DPPH^●^) scavenging activity of the starch samples was determined, using the method of Li et al. [23]. Five mL of starch solution (1 mg/mL) prepared in double deionized water was mixed with 5 mL of 60 mM methanolic DPPH^●^ solution. After incubation in the dark at room temperature (30 min), the absorbance was measured at 517 nm against blank. A control was prepared using methanol instead of the sample. The DPPH^●^ inhibition was acquired by the following formula:(6)$${\mathrm{DPPH}}^{●}\;\mathrm{inhibition}\;\left(\%\right)=[(A_{0}-A_{1})/A_{0}]\times 100$$
where *A*_0_ = absorbance of the control and *A*_1_ = absorbance of the sample.

The proliferation rate of the bifidobacteria (*Bifidobacterium longum* subsp. *longum*) was measured to investigate the prebiotic activity of starch sample [24]. All of the fermentation experiments were conducted in a fermentation medium. The fermentation medium was adjusted to pH 7.0, either with 0.5 M acetic acid or 0.5 M NaOH. The number of bifidobacteria was assessed by measuring the absorbance at 600 nm values in the presence of NK starch, NK starch–lipid complexes or glucose as the carbon source. The carbon source concentration was either 1.25, 2.5, 5, 10, 20, or 40 g/L. The incubation of the bifidobacteria was performed in an anaerobic incubator at 37 °C. The fermentation media’s values for absorbance at 600 nm were examined after 48 h of fermentation.

### 2.4. Statistical Analysis

A completely randomized design was used for the experimental design and three independent experiments were performed. All of the experiments were tested in triplicate and the data were recorded as mean ± standard deviation (SD). A probability value of *p* < 0.05 was deemed significant. Duncan’s multiple range tests were used for the mean comparisons and analysis of variance (ANOVA). SPSS for Windows Version 17.0 (SPSS Inc., Chicago, IL, USA) was used.

## 3. Results and Discussion

### 3.1. Effect of Ultrasonic Condition, FA Concentration, and FA Chain Length on the Starch–FA Complexes Formation Presented by CI and In Vitro Digestibility

#### 3.1.1. Effect of Ultrasonic Time

Figure 1a,b shows the effect of ultrasonic time (0, 5, 10, 15, 30, 45, and 60 min) on the CI value and in vitro digestibility of the NK rice starch in conjunction with the butyric acid (C4:0) compared to the untreated starch. The increases in the CI values of the ultrasonically treated samples were observed with raising ultrasonic time up to 30 min (60.82%) (*p* < 0.05), followed by a dramatic decrease in the CI values until 60 min (Figure 1a), indicating the formation of a higher amylose–lipid complex. The gelatinized starch in the presence of FA (no ultrasonication, 0 min) could result in the transformation of a starch–lipid complex (40%), caused by hydrothermal treatment. The use of ultrasonication could improve the FA incorporation into the amylose helical structure by increasing the FA molecular mobility, as affected by cavitation. This could be due to the ultrasonication dissolving the inflated starch granules [6], allowing more amylose molecules to escape the starch’s inner structure. Moreover, the ultrasonication may improve the ligand dispersibility in gelatinized starch suspensions. The results were consistent with Kang et al. [6], who observed that the ultrasonic treatment for 15 min increased the CI values from 46% to 56% of the corn starch–capric acid (C10:0) complex, when compared to the untreated sample. However, the samples treated ultrasonically for more than 30 min resulted in a reduction in the CI values. The report noted that a rise in the release of the amylose molecules was seen as being aided by the ultrasonic treatment [6], albeit the precise mechanism of the drop in the CI values with a longer sonication duration was unknown. Different investigations have shown different ideal ultrasonic times for the released amylose molecules to form complexes with lipid molecules [6,9].

When increasing the ultrasonic time up to 15 min, there was a noticeable increase in RS accumulation with a decrease in RSD accumulation, however the RS remained constant thereafter with a continuously decreasing RSD (Figure 1b). A slow increase in SDS was detected up to 10 min of ultrasonication, followed by a progressive decrease until 60 min (Figure 1b), indicating the transformation of the digestible starch to indigestible RS. This was confirmed by the CI values of the ultrasonically treated samples with a varying ultrasonic time (Figure 1a).

#### 3.1.2. Effect of Ultrasonic Amplitude

The CI values of the ultrasonically treated NK starch, in conjunction with butyric acid by various ultrasonic amplitudes (0, 10, 20, 30, and 40%), are shown in Figure 1c. The CI values of the ultrasonically aided samples were greater than those of the control (0 min, *p* < 0.05), indicating that ultrasonication enhanced the complexation of starch–butyric acid. An increase in the CI values with increasing ultrasonic amplitude up to 20% was observed, followed by a dramatic decrease in the CI values (*p* < 0.05). This finding indicated that excessive ultrasonic intensity had a negative effect on the molecular inclusion of FA into the amylose helix. This may result in the degradation of the amylose helical structure, or the final complex structure. The observation was similar to the findings of Liu et al. [25], who revealed that low-power-density ultrasound facilitated in the development of the complex, while increasing ultrasonic amplitude steadily decreased the CI values.

Generally, the starch–lipid complex resisted enzymatic digestion [26]. In the ultrasonically treated samples, an increase in the RS content with an increasing ultrasonic amplitude up to 20% was observed, followed by a progressive decrease (*p* < 0.05) (Figure 1d). It was consistent with the CI values (Figure 1c). It was reported that the structure of the starch suspension became more homogeneous with the increase in the ultrasound amplitude and duration [17]. When ultrasonication was used, the swollen starch granules’ double-helix structure was compelled to open, disrupting their natural semicrystalline structural configuration and revealing their interior lamination structure [9]. Additionally, more linear amylose chains were simultaneously liberated from the starch granules’ inner structure and took part in the formation of the FA complex [17]. However, the complex dropped when the ultrasonic amplitude was too high. This decrease was mostly attributed to the violent ultrasound-induced starch-chain disintegration [17].

The reduction in the RDS and SDS amounts was found in response to the changes in RS content (Figure 1d), suggesting that the digestive starch was converted to an indigestible counterpart. The enzymatic digestibility of the ultrasonically treated samples subjected to amplitudes greater than 20% was increased, which was most likely due to the increased occurrence of free amylose molecules caused by an intense ultrasonic cavitation [6,25]. These amylose molecules may be susceptible to enzymatic hydrolysis. According to the findings, the ultrasonic time and amplitude appeared to be crucial variables for facilitating the development of the complexes, which restricts amylolytic activity.

#### 3.1.3. Effect of FA Concentration

Figure 1e depicts the effect of the butyric acid concentrations (1, 2.5, 5, 7.5, and 10%) on the CI values of NK starch–lipid complex produced by 20% ultrasonic amplitude for 30 min. The CI values of the ultrasonically treated samples varied from 60.71 to 60.82% when the butyric acid concentrations were increased from 1% to 5%, with no significant difference (*p* > 0.05). The highest CI value was noticeable (77.18%) at 7.5% butyric acid and decreased to 73.63% when the concentration of butyric acid was increased to 10% (*p* < 0.05).

Increasing the butyric content from 1% to 2.5% had no significant influence on the formation of RS in the ultrasonically treated samples (*p* > 0.05; Figure 1f). When the butyric acid concentration was increased from 5% to 7.5%, there was an increase in the RS concentration in the treated samples, followed by a slight decrease in the RS content at 10% butyric acid (*p* < 0.05; Figure 1f). Thus, the formation of a starch–lipid complex resulted in the formation of RS. The synthesis of the starch–lipid complex with a low amount of free FA (1–2.5%) resulted in the transformation of RDS to SDS with the negligible formation of RS (Figure 1f). This was most likely caused by an inappropriate starch to lipid ratio, which resulted in a low concentration of newly formed RS. The greater extent of RS formation could be obtained by the synthesis at high free-FA concentrations (5–7.5%), resulting in the conversion of RDS or SDS to RS. However, a decrease in RS content was detected at 10% butyric acid, due to less amylose leaching from the starch granule during the thermal gelatinization. As a result, the establishment of lower amylose–lipid complexes took place. According to Chang et al. [26], an increased FA content can coat the granule surface, associate with amylose, and impair water mobility into the granules, leading to suppression of the starch gelatinization. As a consequence, using 7.5% free FA resulted in the highest CI value and RS content of the NK starch–lipid complex.

#### 3.1.4. Effect of FA Chain Length

The effects of the FA chain lengths on the CI values of the ultrasonically aided NK starch–lipid complexes prepared by 7.5% lipid at 20% amplitude for 30 min are depicted in Figure 1g. The decrease in the CI values from 77.11% to 39.68% was caused by increasing the length of the saturated FA chain from 4 to 18 carbons (*p* < 0.05). However, after the addition of polyunsaturated linoleic acid (C18:2) to the starch molecules, the CI value increased to 65.85%. Overall, the inclusion of the short chain butyric acid (C4:0) resulted in the highest CI value (*p* < 0.05), whereas the CI values were dramatically reduced when complexing with medium- (C12:0) or long-chain (C18:0) saturated FA into starch molecules, and significantly increased when complexing with the linoleic acid (C18:2) counterpart. The butyric and linoleic acids are naturally liquid FA, whereas the lauric and stearic acids are solid FA. The inclusion of the liquid FA could be more deeply embedded into the amylose molecules than the solid FA. This could be owing to the various FA states’ differing dispersibilities. The solid lipid dispersed poorly in the starch suspension compared to the liquid lipid [6], resulting in a poor filling up into the amylose helix. This finding was in line with that of Thakur et al. [27], who found that the CI values reduced as the FA chain length increased, indicating a stronger role for short-chain FA in complex formation. The shorter chain FA were thought to be more disseminated in the gelatinized mixture, allowing for easier associations with the amylose and strengthening the CI complex. Similarly, Kawai et al. [28] found that, as the content of the carbons in the FA increased from 4 to 18, the CI values tended to decrease (77.11% to 39.68%). Furthermore, the CI values increased with the value of the double bonds in the FA (C18:0; CI = 39.68% and C18:2, CI = 65.38%) [20]. Because the double bonds bend the carbon chain of an unsaturated FA, the amount of carbons used in the complex transformation appears to be lower than that of a saturated FA with a similar carbon-chain length. Wang et al. [29] reported that the CI value of FA reduced with the increasing chain length, with significant differences observed among the saturated FA. It should be noted that the liquid short-chain FA (C4:0) complexed into the amylose helix more than the liquid long-chain unsaturated linoleic acid (C18:2). This result could be explained by the fact that butyric acid’s carbon chain is shorter (with a better solubility in water) and simpler to liberate from the hydrophobic core of amylose than that of linoleic acid, which had a low water dispersibility. The short chain butyric acid was more water miscible than the unsaturated long-chain linoleic acid, which conjugated more often with the host molecules.

NK starch–butyric acid complex had the highest RS content of 80.78% (*p* < 0.05), followed by the starch–linoleic acid complex (69.12%), the starch–lauric acid complex (51.62%), and the starch–stearic acid complex (48.93%) (Figure 1h), respectively. These were noticed to be highly correlated with the CI values of the different FA–starch complexes (Figure 1g). The fluctuation of the RDS and SDS contents as influenced by the FA chain length (Figure 1h) may be attributed to the transformation of RDS to SDS or RS, as well as the conversion of SDS to RS. The crystalline regions in RS are more compactly ordered than the amorphous regions, making them less particularly susceptible to amylolytic enzyme attack [26,27]. The granules of starch are typically spherical in shape and feature polycrystalline structures with an amorphous area and crystalline structure. The starch molecules are disorganized in the amorphous zone, but they are precisely arranged in double helices in the crystalline region [24].

### 3.2. Physico-Chemical Properties

#### 3.2.1. Water Absorption Capacity (WAC) and Oil Absorption Capacity (OAC)

Table 1 shows the WAC and OAC of native NK starch and lipid-treated starches. When compared to the native starch (1.73 g/g), there was a substantial increase in WAC of the starch–lipid complexes (*p* < 0.05), ranging from 4.27–4.6 g/g depending on the integrated FA. The increased WAC was mostly attributable to the heated gelatinization and intensified ultrasonic treatment, which increased the exposure of the hydroxyl groups in amylose and amylopectin. These data were consistent with the previous study of Ashwar et al. [22], who found that the rice starch subjected to a dual autoclaving-retrogradation treatment had a higher WAC than the native rice starch. When compared to the native counterpart (2.19 g/g), the starch–lipid complexes had a lower OAC (1.85–1.95 g/g, Table 1). The reduced OAC of the complexes would be owing to the intact amylose–amylose and/or amylose–amylopectin interactions as a result of high temperature/ultrasonication, or to the unavailable oil integration in a porous starch network of amylose or amylopectin caused by the establishment of a starch–lipid complex. This result was supported by the lower OAC of the complexes with higher CI values with improved WAC, such as starch–butyric and starch–linoleic complexes (Figure 1h).

#### 3.2.2. Swelling and Solubility Indices

The swelling and solubility index of the native NK starch and starch–lipid complexes are shown in Table 1. The swelling index varied between 2.70–7.54 g/g for native NK starch, 9.51–17.41 g/g for NK starch–butyric acid, 7.45–13.59 g/g for NK starch–lauric acid, 5.72–14.56 g/g for NK starch–stearic acid, and 9.67–22.50 g/g for NK starch–linoleic acid complex within the temperature range from 60 to 90 °C. Overall, the incorporation of free FA into the starch molecules by ultrasonication increased the swelling index. This result was consistent with the findings of Wang et al. [30], who revealed that the swelling power of potato starch was improved with the addition of FA when compared to native potato starch. Typically, the swelling power of the starch–lipid complexes varied depending on the source of the starch and the type of the lipids [29]. As a result, the differences in the treated starch swelling-capacity may be linked to the modification procedures. The hydrothermal treatment, followed by the ultrasonication employed in this study, may have resulted in a diminished, ordered structure and an association between amylose–amylose or amylose–amylopectin in starch granules, resulting in increased water diffusion and swelling power. An integrated, unsaturated linoleic acid into starch molecule showed the greatest swelling index in all of the tested temperatures (*p* < 0.05), followed by short-chain butyric acid, medium-chain lauric acid, and long-chain stearic acid, respectively. In addition, the swelling index of starch–lipid complexes and native NK starch increased significantly as the temperature rose (*p* < 0.05), indicating that the swelling index is temperature dependent [22].

The solubility index of all of the ultrasonically constructed starch–lipid complexes was slightly increased when compared to the native counterpart (*p* < 0.05, Table 1). The intense cavitation formed during ultrasonication may hasten the amylose leaching, which could be attributed to the higher solubility of the ultrasonically prepared samples. It should be noted that the complexes with higher CI values, such as starch–butyric acid and starch–linoleic acid, had a higher solubility index (Table 1). Wang et al. [30] noticed that the CI of starch–FA was related to solubility and swelling power. The higher the CI, the greater the solubility and swelling power [31]. At 90 °C, where the bulk of the starch granules were swollen and gelatinized, all of the starches had the highest solubility (Table 1).

#### 3.2.3. Light Transmittance (LT)

The impact of cold storage on the LT of the native NK starch and the ultrasonically treated starch–lipid complexes is given in Table 1. The clarity of the starch paste can be determined by measuring LT, which indicates the retrogradation process. The percentage of light transmission in all of the starch samples steadily declined during the whole storage period of 120 h (*p* < 0.05). A higher light transmission was observed in all of the starch–lipid complexes compared to the native starch paste at all of the time points over 120 h (*p* < 0.05), indicating a lower extent of retrogradation. Thus, the ultrasonically treated starch–lipid complex had less starch retrogradation than the native starch, due to the occurrence of the amylose in the starch–lipid complex. The retrogradation process could not completely take place, because the free amylose molecules were not fully available. This result was in agreement with the study of Ashwar et al. [22], who found that the LT of the native and dual autoclave-retrogradation-treated rice starch pastes was reduced, whereas the treated starch pastes had a higher LT than the native starch gels over 120 h of storage periods. The retrogradation of the starch pastes may be linked to a decrease in LT, as the storage time increases [32].

### 3.3. Pasting Behaviors

The RVA pasting characteristics of the native NK starch and starch–lipid complexes are depicted in Table 2. The inclusion of FA into the starch molecules significantly decreased the peak viscosity (*p* < 0.05), compared to the native starch, which varied depending on the types of FA (846.67 to 3295 cP). The prevalent amylose–lipid interactions may also be related to lower water binding to the amylose molecules, resulting in reduced viscosity for a complex, such as lower peak viscosity. The peak viscosity was significantly reduced (*p* < 0.05) as the chain length of the integrated saturated FA increased. This finding was in line with Wang et al. [29], who observed that the peak viscosities of wheat starch–FA samples were lower than those of the corresponding native wheat starch samples as the chain length of the incorporated FA increased. The addition of linoleic acid to the starch molecule resulted in a higher peak viscosity than the lauric and stearic counterparts (*p* < 0.05). The order of peak viscosity among the ultrasonically made starch–lipid complexes was highly related to their swelling index (Table 1).

The peak viscosity of the starchy component was impacted by the degree of amylose leaching, the production of complexes, granular swelling, the friction among the swollen granules, and competitive pressure for free water between the leached amylose and ungelatinized granules [33]. The trough viscosity, breakdown viscosity, setback viscosity, and final peak viscosity of the native NK starch were also higher than those of the starch–lipid complexes (*p* < 0.05; Table 2). The trough viscosity, breakdown viscosity, setback viscosity, and final peak viscosity of the starch–lipid complexes were in the same order as the peak viscosity (Table 2). Furthermore, the final peak viscosity of those lipid complexes increased when compared to the first peak viscosity. The starch–stearic acid complex with the lowest CI value (Figure 1g) had the lowest final peak viscosity (*p* < 0.05), which could be attributed to the steric hindrance power of the C18 chain. Long chain FA, in particular, needs more amylose space to complex, resulting in a less complex formation. According to Wang et al. [30], only a part of the C18 sections penetrate the amylose ring system, leaving the rest exposed to the environment. This prevented the water from entering the starch granules and significantly inhibited starch gelatinization. The short chain butyric acid and bent double-bonds of linoleic acid may fit perfectly inside the amylose helix, burying the lipid hydrophobic moiety within the amylose structure. As a result, greater water absorption took place, leading to higher peak viscosity than other starch–lipid complexes. The native NK starch had significantly higher pasting temperature than that of the starch–lipid complexes (*p* < 0.05), except for the starch–stearic acid complex (90.5 °C). The high melting point of stearic acid may contribute to the higher pasting temperature of its starch complex when compared to other starch–FA complexes. Kaur and Singh [31] reported that cooking rice flour with stearic acid resulted in a greater pasting temperature than those with myristic and palmitic acids.

### 3.4. Thermal Properties

The T_o_, T_p_, T_end_, and ΔH, reflecting the change in the thermodynamic properties of the native NK starch and NK starch–lipid complexes, are presented in Table 2 and Figure 2. The T_o_ of native NK starch, starch–butyric acid, starch–lauric acid, starch–stearic acid, and starch–linoleic acid were 62.51, 96.8, 95.57, 99.82, and 100.48 °C, respectively, indicating that the formation of a starch–lipid complex can raise the first melting temperature. The T_p_ values for the corresponding samples were 179.54, 125.48, 125.50, 123.04, and 126.28 °C, while the T_end_ values were 179.54, 125.48, 125.50, 123.04, and 126.28 °C, respectively (Table 2). The endothermic peak of the starch–lipid complexes was sharper than that of the native starch, indicating a higher degree of ordered structure (Figure 2). The formation of the starch–lipid complexes, along with the retrograded amylose crystallites, may contribute to more heat-stable complexes than that of the native starch. There was no statistically significant difference in the melting temperature as the length of the incorporated FA chain into the amylose molecule increased (Table 2) (*p* > 0.05), demonstrating the similar ordered formation as influenced by the ultrasonication. The melting temperature of the complexes was found to be slightly lower than that of the native starch (*p* < 0.05). The endothermic peak of native starch, on the other hand, was largely broad, making it difficult to determine the precise melting point. Furthermore, the increasing ΔH of the starch–lipid complexes was observed, ranging from 528.28–718.54 J/g when compared to native starch (84.96 J/g). Among the starch–lipid samples, ΔH were greater in the following order: linoleic acid > lauric acid > butyric acid > stearic acid, indicating a lower degree of V-type complex formation for the solid long-chain stearic acid. This was highly agreed upon by the CI value (Figure 1g). According to Zhang et al. [34], the starch–solid lipid complexes had a lower melting enthalpy than the liquid lipid, leading to a low complexation capability to enroll in the V-type crystalline structure. The melting enthalpy of ultrasonically treated corn starch–lipid complexes decreased with the rising carbon numbers (C10–C14), indicating a lesser degree of V-type complex for the long chain FA [6]. The melting enthalpy of starch–lipid complexes may be influenced by the increased amount of linear amylose molecules involved in complexation. The lipid should be present in the starch dispersion as a guest molecule, in order to interact with the amylose molecules, which affects the lipid solubility in complexation with host molecules. As a consequence, ultrasonication plays a critical role in the formation of the V-type crystalline complexes [6].

### 3.5. XRD Pattern

Figure 3 depicts the X-ray diffraction spectra of the native NK starch and starch–lipid complexes. The native NK starch exhibited an A-type diffraction pattern with prominent peaks at diffraction angle (2θ) of 14.2, 17.37, 18.7, and 23.37°. The peaks at 21.3 and 24.20° were identified as the crystalline pattern of free FA. The peak intensities of amylose–FA complexes and free FA decreased and significantly increased as the carbon-chain length increased [29]. The double helices were packed tightly, with only four inter-helical water molecules, and the center of the lattice contained a pair of double helices [35]. The crystalline structure of the starch was transformed from the A-type to the V-type pattern when the NK starch–FA mixture was gelatinized (90 °C for 30 min) and ultrasonicated. The 20° peak was attributed to a well-formed V-type structure, which was assigned to the tightly packed single amylose helices complexed with various polar and non-polar compounds, such as FA [5]. The V pattern is typically defined as a single, left-handed helical structure with an inclusion complex present in the helical channel [36]. There was no amylopectin diffraction peak (2θ of 17°) in the NK starch–FA complex diffraction pattern, indicating that the amylose–lipid complex inhibited the amylopectin recrystallization [6]. The diffraction intensity of the ultrasonically treated NK starch–lipid complexes was higher than that of the native starch. This behavior revealed that ultrasonication could promote the formation of starch–lipid complexes. The shear force and cavitation potential of the ultrasonication disintegrated the granules, resulting in the formation of more linear amylose units. As a result, the likelihood of the linear amylose interacting with the lipids increased.

### 3.6. FTIR Spectra

The FTIR spectra of the native NK starch and starch–lipid complexes are shown in Figure 4. The stretching vibrations of poly–OH and C–H, as well as the bending vibrations of O–H, were typically attributed to the bands at 3295, 2927, and 1240 cm^−1^, respectively. The cross-link of the hydrogen bonds between the carbonyl groups of the FA and the hydroxyl moieties of amylose would be responsible for the shift of the peak at 3300 cm^−1^. The NK native rice starch had a wider peak at 2927 cm^−1^, which was due to the stretching vibration peak of C–H (Figure 4). As the FA were added to the NK rice starch, the two new peaks at 2927 and 2850 cm^−1^ were seen. This could correspond to the asymmetric and symmetric stretching of CH_2_ and CH_3_ of FA. The bands in the region of 1365–1413 cm^−1^ were matched for the C–H bending vibrations. The C–O–H bending was observed at 1047 and 1022 cm^−1^, which corresponded to the crystalline and amorphous structures of starch, respectively [22]. The characteristic peak of FA could be seen around 1700 cm^−1^, which was due to the absorbance of the carbonyl group. The complexed FA showed a higher peak intensity of the carbonyl group at 1700 cm^−1^ (Figure 4). In comparison to native starch, the FTIR spectra of the starch–lipid complexes revealed one extra band at 1709 cm^−1^ (Figure 4). When FA was added to an amylose molecule, the peak attributed to the carbonyl group of the free FA shifted. The same trend was reported by Wang et al. [29], who found the carbonyl band at 1709 cm^−1^. This band could be attributed to the breaking of the hydrogen bonds between the carboxyl groups of FA in the crystalline state, indicating FA incorporation inside the amylose helix. Since FA enter the amylose double helix and form hydrogen bonds with starch molecules, no absorption peak of FA at 720 cm^−1^ was observed in the complexes (Figure 4). The additional band around 2356 cm^−1^ was also found in the newly formed complexes.

### 3.7. SEM

The morphologies of the native starch and ultrasonically treated starch–lipid complexes are displayed in Figure 5. In the native starch, the smooth surfaces of the integrity starch granules could be visualized. The ultrasonically treated lipid complexes had a distinct morphological structure, with a greater number of flaky and grooved shapes, deeper layered strips, and a rather uneven granule surface. The appearance of disrupted granules in the starch–lipid complexes may be caused by partial gelatinization and ultrasonically cavitation. Ding et al. [14] found that the ultrasound facilitated interactions between trimmed chains, particularly amylose residues, during recrystallization, resulting in the formation of agglomerated starch–lipid complex granules. The oddly shaped particles on the surfaces of the swollen NK rice starch granules were attributed to amylose leaching, which may have contributed to the development of a complex layer on the granule surface. Shi et al. [37] observed that the FA could not permeate the starch granules, but instead formed a complex coating around the starch granule. Depending on the fatty acid employed, the starch–lipid complex had a different morphology. In contrast to the smaller butyric acid (Figure 5c,d), which did not render the starch–lipid complexes with the coated layer in the granules, lauric (Figure 5e,f), stearic (Figure 5g,h), and linoleic (Figure 5i,j) acids did.

### 3.8. In Vitro Bioactivities of Starch–FA Complexes

#### 3.8.1. Bile Acid Binding Capacity

The bile acid-binding capacity indicates cholesterol-lowering activity by increasing the fecal bile acid excretion [22]. Table 3 shows the bile acid-binding capacities of native NK starch and ultrasonically treated NK starch with various FA. The NK starch–lipid complexes showed significantly higher bile acid-binding capacities (12.64–15.21%) than that of the native sample (12.02%). Simsek and El [38] revealed that taro RS had a significantly higher bile acid-binding capacity (7.6%) than the native taro starch (5.2%). As a result, the higher RS content of starch–lipid complexes (Figure 1h) may be associated with a greater bile acid binding capacity when compared to the native NK starch. The increased length of the saturated FA chain led to an increase in bile acid-binding capacity (Table 3), which was unrelated to the amount of RS content (Figure 1h). The linoleic acid inclusion in the starch molecules resulted in the lowest bile acid-binding capacity among the starch–lipid complexes (Table 3), despite having a higher RS content than the starch–stearic acid complex. This might be because, in comparison to the more ordered starch–butyric acid or starch–linoleic acid complexes, the ultrasonically treated starch–stearic acid complex has a more irregular structure and more branches, making it simpler to form an emulsion with the bile acid molecules. Typically, bile acids solubilize cholesterol in mixed micelles with phospholipids. The bile acids serve to emulsify fats and aid in their digestion after they are secreted into the intestine [39]. The ability of RS to form inclusion complexes with bile acid was strongly correlated with its double helical structure, suggesting that this is the alternative route through which bile acid may be trapped [40]. According to Abadie et al. [41], the hydrophobic interior of the helical structures, generated by the 1,4-linked glucose units of amylose and amylopectin, is the most plausible binding site for an amphiphilic ligand with RS. Additionally, the stability of the starch helical complex and the number of binding sites per unit both affect how refined this interaction is [41]. In essence, the presence of affine lipid molecules causes the starch–lipid complex to have a greater hydrophobic area. The different FA subtypes, though, might provide the complexes with varying degrees of hydrophobicity. According to the findings, the bile acid-binding capability differed, depending on the type of FA. According to the reports, the RS samples with rather loose interior structures may also be responsible for the most notable improvement in RS’s ability to bind bile acids. This could be explained by the hypothesis put forth by Zhou et al. [40], that bile acids may penetrate through the surface fissures of the starch–stearic acid complex, leading to a greater interaction between the RS and bile acids. In comparison to native NK starch, starch–lipid complexes have a greater health-promoting effect, because of their potential cholesterol-lowering ability.

#### 3.8.2. Radical Scavenging Activity

The DPPH radical scavenging-activity of native starch and starch–lipid complexes is presented in Table 3, which is mainly due to the presence of the hydroxyl groups of the starch molecule. The DPPH radical scavenging-activity of starch–FA complexes was higher than that of native starch (*p* < 0.05), and varied depending on the type of FA. The explanation for this could be related to amylose and FA’s single helical structure, which exposes more hydroxyl groups and contributes to their higher antioxidant activity [23], as influenced by thermal gelatinization and ultrasonication. The greater exposed hydroxyl groups of starch effectively scavenged more of the DPPH free radicals, resulting in the termination of the free radical lipid-chain-reaction [22]. Among the starch–lipid complexes, the complex with lauric acid had the highest DPPH radical scavenging activity of 54.74% (*p* < 0.05), which might be due to the potential radical scavenging activity of the medium-chain lauric acid.

#### 3.8.3. Proliferation of Bifidobacteria

The starch–lipid complex, having RS as dietary fiber, promotes the growth of bifidobacteria, which resists enzymatic digestion in the small intestine and enters the colon with functions comparable to the non-starch polysaccharide fibers as prebiotic activity. The proliferation of the bifidobacteria in the NK starch–lipid complexes with different FA are shown in Table 3, which observed the absorbance at 600 nm (OD_600_) as an indicator of bifidobacteria growth. The increased OD_600_ values were observed with increasing concentrations of the native starch and starch–lipid complexes up to 20 g/L (*p* < 0.05), demonstrating the accelerative potential of bifidobacterial growth. This outcome was very close to Zheng et al. [2], who found that the starch–oleic acid complex can facilitate the growth and reproduction of enteric bacteria as butyrate-producer, particularly Bifidobacterium, in rats fed a high-fat diet when compared to a heat-moisture treatment-rice starch-fed diet. However, when the concentrations reached 40 g/L, there was a slight decrease in the OD_600_ value in all of the treatments, indicating an adverse effect on microbial proliferation. This circumstance could be attributed to high carbon source concentrations in the medium, which resulted in high osmotic pressure and the accumulation of fermentation products, both of which impeded bifidobacterial growth. All of the starch–lipid complexes showed comparable or even higher bifidobacterial growth than the positive control glucose (Table 3), indicating prebiotic activity. It should be noted that the bifidobacterial growth stimulation-ability of the starch–lipid complex was governed by both types of integrated FA and the complexed concentration (Table 3), in which the starch–linoleic acid, starch–butyric acid, and starch–stearic acid complexes showed the greater stimulation of bifidobacterial growth at a low concentration of 1.25–5 g/L. The starch–lauric acid complex, on the other hand, facilitated bifidobacterial proliferation at high concentrations of 10–20 g/L, whereas the starch–stearic acid complex resulted in a dramatic decrease in the microbial numbers. According to Zhou et al. [42], the gut microbiota and their metabolized products differed significantly, depending on the type of saturated FA complexed in the high-amylose maize starches fermented with human fecal innocula at different fermentation stages. Overall, the starch–butyric acid and starch–linoleic acid complexes outperformed other complexes in terms of bifidobacterial proliferation, which was strongly associated with the high RS content (Figure 1h). Furthermore, the native NK starch exhibits naturally prebiotic activity.

## 4. Conclusions

Ultrasonication promoted the production of starch–lipid complexes with higher CI values and RS contents, as well as superior in vitro bioactivities, that could be used for functional food. The CI and in vitro digestibility of the ultrasonically prepared NK starch–FA complex were varied, depending on the ultrasonic conditions as well as the concentration and type of integrated FA, with 7.5% butyric acid at 20% amplitude for 30 min ultrasonication being the optimum conditions. DSC, FTIR, and XRD measurements revealed that the NK starch–lipid complexes formed V-type complexes. Improved bile acid binding-activity, DPPH radical scavenging capacity, and bifidobacterial proliferation were observed in the NK starch–lipid complex. However, it is necessary to conduct human subject studies to verify the starch–lipid complex’s beneficial qualities. In order to fulfill the practical application on an industrial scale, it is also necessary to investigate the applicability in food products.

## Figures and Tables

**Figure 1 foods-11-02430-f001:**
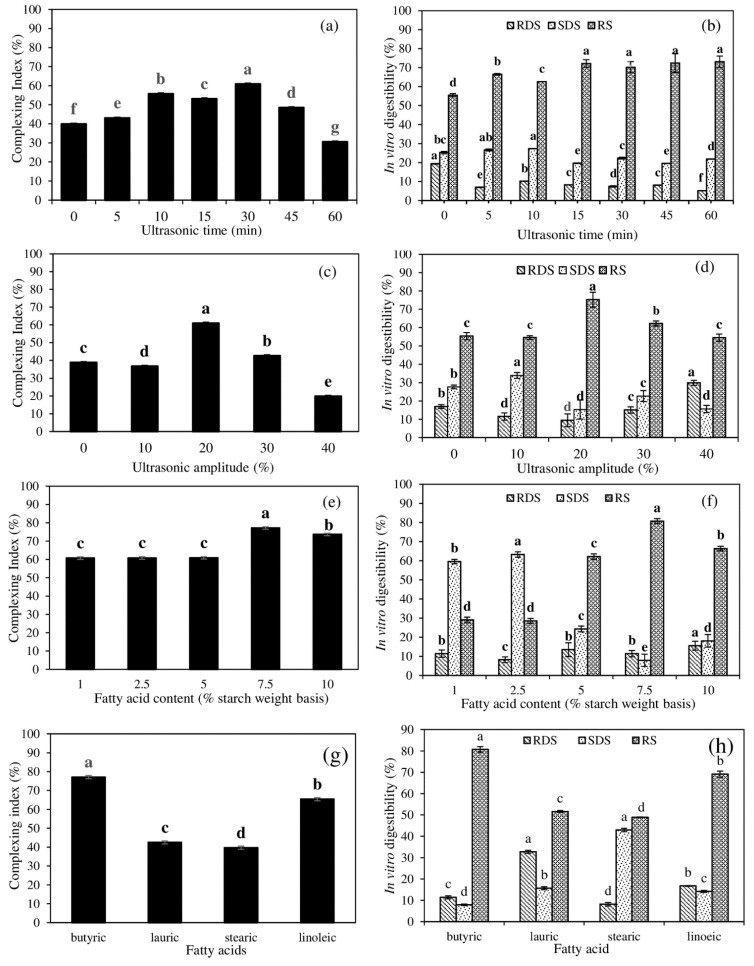
The effect of synthesized conditions on the complexing index (**a**,**c**,**e**,**g**) and in vitro digestibility (**b**,**d**,**f**,**h**) of the ultrasonically treated *Noui Khuea* (NK) starch–lipid complex. Effect of ultrasonic time (**a**,**b**); effect of ultrasonic amplitude (**c**,**d**); effect of fatty acid concentration (**e**,**f**); effect of fatty acid chain length (**g**,**h**). The reaction was conducted for 30 min to determine the impact of ultrasonic amplitude, fatty acid concentration, and fatty acid chain length. Bars represent the standard deviations from triplicate determinations. Different letters under the same parameter indicate the significant differences (*p* < 0.05). RDS, SDS, and RS were analyzed separately. RDS = rapidly digestible starch; SDS = slowly digestible starch; RS = resistant starch.

**Figure 2 foods-11-02430-f002:**
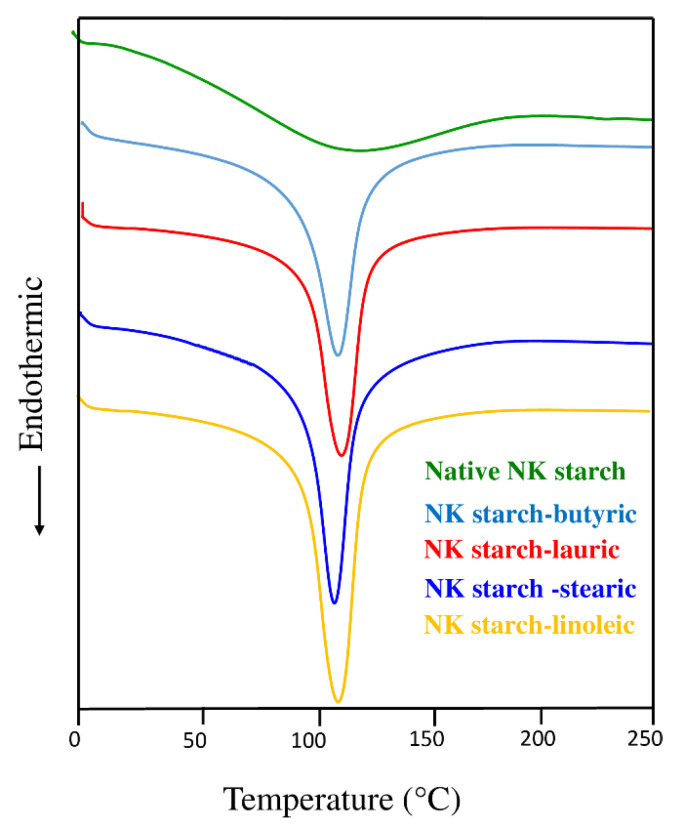
Differential scanning calorimeter (DSC) thermogram of native *Noui Khuea* (NK) starch and NK starch–lipid complexes with different fatty acid chain length.

**Figure 3 foods-11-02430-f003:**
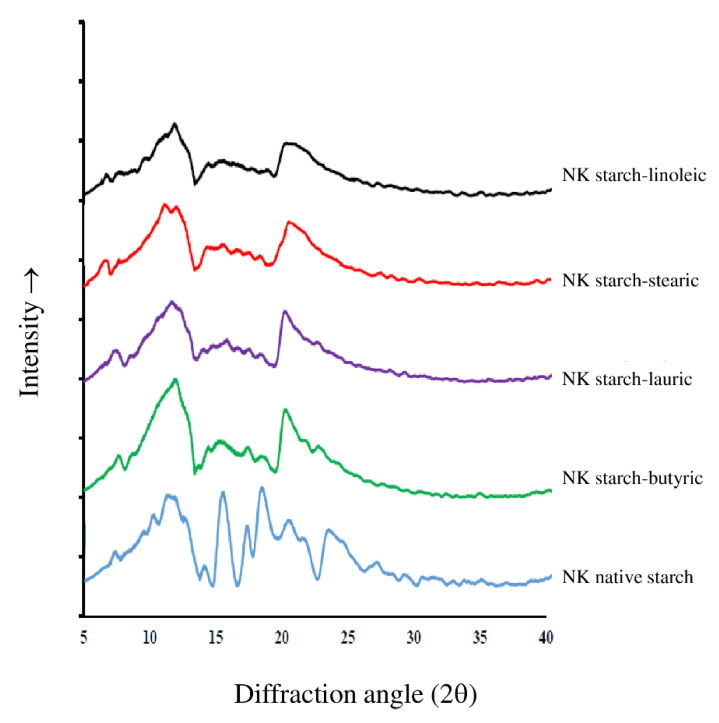
X-ray diffraction (XRD) pattern of native *Noui Khuea* (NK) starch and NK starch–lipid complexes with different fatty acid chain length.

**Figure 4 foods-11-02430-f004:**
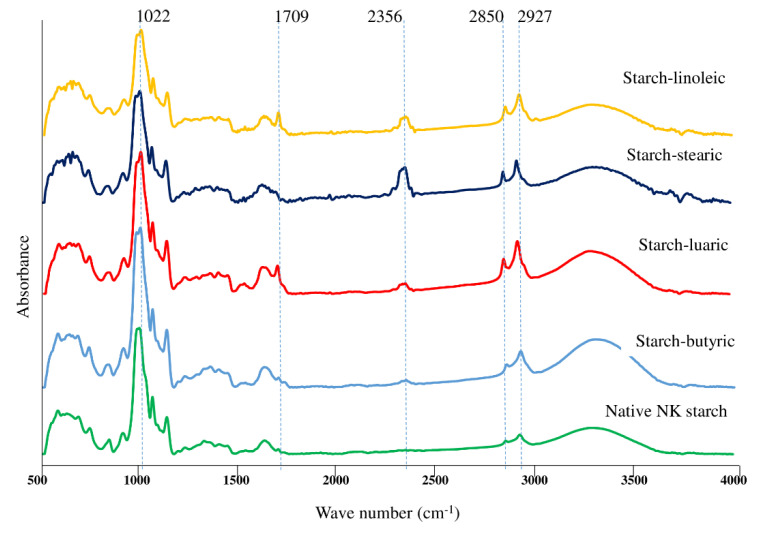
Fourier-Transform infrared (FTIR) spectra of native *Noui Khuea* (NK) starch and NK starch–lipid complexes with different fatty acid chain length.

**Figure 5 foods-11-02430-f005:**
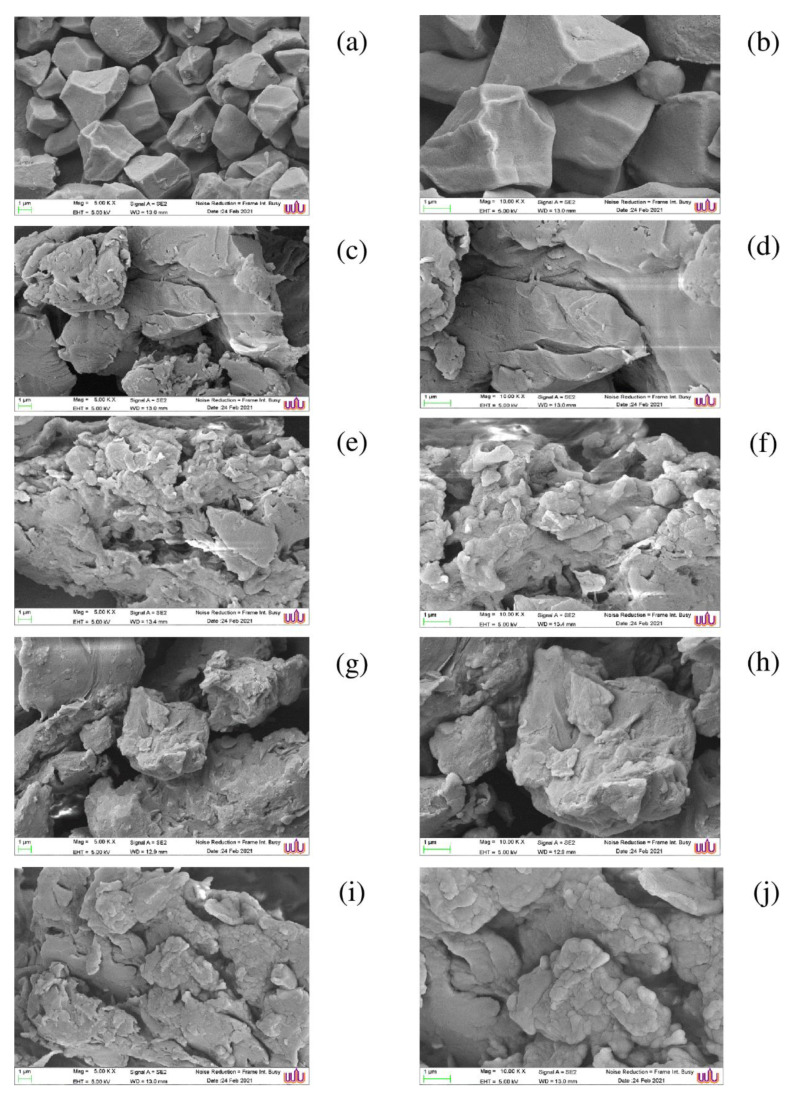
Scanning electron microscopic (SEM) images of native *Noui Khuea* (NK) starch and starch–lipid complexes with different fatty acid chain length. Native NK starch (**a**,**b**); starch–butyric acid complex (**c**,**d**); starch–lauric acid complex (**e**,**f**); starch–stearic acid complex (**g**,**h**); starch–linoleic acid complex (**i**,**j**). Magnification: 5000× (**a**,**c**,**e**,**g**,**i**) and 10,000 × (**b**,**d**,**f**,**h**,**j**). EHT: 10 kV.

**Table 1 foods-11-02430-t001:** Physico-chemical properties of native *Noui Khuea* (NK) starch and NK starch–lipid complexes with varying fatty acid chain length.

Parameter	Native NK	NK–Butyric	NK–Lauric	NK–Stearic	NK–Linoleic
Water absorption capacity (g/g)	1.73 ± 0.03 ^d^	4.35 ± 0.01 ^b^	4.42 ± 0.07 ^b^	4.27 ± 0.02 ^c^	4.57 ± 0.01 ^a^
Oil absorption capacity (g/g)	2.19 ± 0.09 ^a^	1.85 ± 0.02 ^e^	1.95 ± 0.001 ^b^	1.93 ± 0.01 ^c^	1.90 ± 0.03 ^d^
Swelling index (g/g)					
60 °C	1.70 ± 0.60 ^e^	8.51 ± 0.07 ^b^	6.45 ± 0.09 ^c^	4.72 ± 0.05 ^d^	8.67 ± 0.11 ^a^
70 °C	2.17 ± 0.06 ^e^	10.65 ± 0.08 ^b^	7.28 ± 0.08 ^c^	5.50 ± 0.10 ^d^	15.55 ± 0.09 ^a^
80 °C	4.47 ± 0.09 ^e^	12.88 ± 0.11 ^b^	10.65 ± 0.13 ^c^	7.90 ± 0.11 ^d^	19.30 ± 0.24 ^a^
90 °C	6.54 ± 0.08 ^e^	16.41 ± 0.06 ^b^	12.59 ± 0.16 ^d^	13.56 ± 0.08 ^c^	21.50 ± 0.22 ^a^
Solubility index (g/g)					
60 °C	0.55 ± 0.01 ^d^	0.73 ± 0.01 ^a^	0.58 ± 0.01 ^c^	0.65 ± 0.02 ^b^	0.65 ± 0.04 ^b^
70 °C	0.57 ± 0.01 ^e^	0.78 ± 0.01 ^b^	0.59 ± 0.01 ^d^	0.67 ± 0.01 ^c^	0.82 ± 0.01 ^a^
80 °C	0.57 ± 0.01 ^e^	0.79 ± 0.01 ^b^	0.65 ± 0.01 ^d^	0.71 ± 0.01 ^c^	0.85 ± 0.01 ^a^
90 °C	0.59 ± 0.01 ^e^	0.85 ± 0.01 ^b^	0.67 ± 0.01 ^d^	0.73 ± 0.01 ^c^	0.87 ± 0.01 ^a^
Light transmittance (%)					
0 h	5.68 ± 0.03 ^e^	28.06 ± 0.05 ^b^	10.24 ± 0.07 ^c^	7.62 ± 0.06 ^d^	29.75 ± 0.15 ^a^
24 h	3.66 ± 0.02 ^e^	26.44 ± 0.08 ^a^	7.46 ± 0.19 ^c^	4.17 ± 0.04 ^d^	25.54 ± 0.18 ^b^
48 h	2.71 ± 0.01 ^d^	23.89 ± 0.07 ^a^	5.08 ± 0.14 ^c^	2.70 ± 0.07 ^d^	22.69 ± 0.09 ^b^
72 h	1.97 ± 0.01 ^d^	16.61 ± 0.07 ^b^	4.22 ± 0.04 ^c^	2.04 ± 0.04 ^d^	21.32 ± 0.23 ^a^
96 h	0.63 ± 0.02 ^d^	10.48 ± 0.18 ^a^	1.28 ± 0.04 ^c^	1.55 ± 0.08 ^b^	10.65 ± 0.12 ^a^
120 h	0.21 ± 0.01 ^e^	5.36 ± 0.02 ^a^	1.08 ± 0.03 ^c^	0.47 ± 0.02 ^d^	4.61 ± 0.05 ^b^

Values are given as mean ± standard deviation from triplicate determinations. Different letters in the same row indicate significant differences (*p* < 0.05). The NK starch was subjected to a 30-min ultrasonic reaction with 7.5% fatty acid at 20% amplitude.

**Table 2 foods-11-02430-t002:** Pasting and thermal properties of native *Noui Khuea* (NK) starch and NK starch–lipid complexes with varying fatty acid chain length.

Parameter	Native NK	NK–Butyric	NK–LAURIC	NK–Stearic	NK–Linoleic
RVA pasting properties					
Peak viscosity (cP)	4222 ± 19 ^a^	3295 ± 76 ^b^	2275 ± 10 ^d^	847 ± 24 ^e^	2787 ± 34 ^c^
Trough viscosity (cP)	3128 ± 62 ^a^	2802 ± 23 ^b^	1736 ± 9 ^d^	604 ± 10 ^e^	2324 ± 22 ^c^
Breakdown viscosity (cP)	1095 ± 87 ^a^	493 ± 54 ^b^	539 ± 7 ^b^	243 ± 14 ^c^	463 ± 13 ^b^
Final viscosity (cP)	6660 ± 60 ^a^	3492 ± 55 ^b^	2084 ± 36 ^d^	641 ± 19 ^e^	2950 ± 44 ^c^
Setback viscosity (cP)	3532 ± 73 ^a^	710 ± 55 ^b^	349 ± 20 ^c^	37 ± 9 ^d^	626 ± 26 ^b^
Pasting temperature (°C)	82.10 ± 1.8 ^b^	50.36 ± 0.1 ^d^	66.63 ± 0.25 ^c^	90.50 ± 3.3 ^a^	50.31 ± 0.2 ^d^
Thermal property					
Onset temperature; T_o_ (°C)	62.51 ± 5.2 ^d^	96.80 ± 2.62 ^c^	95.57 ± 2.8 ^c^	99.82 ± 1.71 ^b^	100.48 ± 1.2 ^a^
Endothermic peak; T_p_ (°C)	120.33 ± 4.2 ^a^	113.6 ± 1.4 ^b^	113.93 ± 0.8 ^b^	112.64 ± 1.6 ^b^	115.13 ± 1.7 ^b^
End temperature; T_end_ (°C)	179.54 ± 6.1 ^a^	125.48 ± 1.12 ^b^	125.50 ± 0.8 ^b^	123.04 ± 0.78 ^c^	126.28 ± 1.2 ^b^
Enthalpy; ΔH (J/g)	84.96 ± 6.3 ^e^	616.30 ± 5.34 ^c^	637.27 ± 5.7 ^b^	528.82 ± 4.6 ^d^	718.53 ± 4.5 ^a^

Values are given as mean ± standard deviation from triplicate determinations. Different letters in the same row indicate significant differences (*p* < 0.05).

**Table 3 foods-11-02430-t003:** In vitro bile acid binding capacity, free radical scavenging activity, and Bifidobacterium cell stimulation of native *Noui Khuea* (NK) starch and NK starch–lipid complexes with varying fatty acid chain length.

Parameter	Native NK	NK–Butyric	NK–Lauric	NK–Stearic	NK–Linoleic
Bile acid-binding capacity (%)	12.02 ± 0.01 ^e^	12.98 ± 0.01 ^c^	14.23 ± 0.25 ^b^	15.21 ± 0.08 ^a^	12.64 ± 0.07 ^d^
DPPH radical scavenging activity (%)	41.78 ± 0.08 ^e^	48.06 ± 0.11 ^b^	54.74 ± 0.05 ^a^	45.12 ± 0.40 ^d^	47.21 ± 0.16 ^c^
Bifidobacterium cell (OD_600_ nm) *					
1.25 g/L	0.05 ± 0.01 ^b^	0.08 ± 0.02 ^a^	0.04 ± 0.01 ^b^	0.10 ± 0.01 ^a^	0.04 ± 0.01 ^b^
2.5 g/L	0.09 ± 0.01 ^c^	0.12 ± 0.02 ^b^	0.06 ± 0.02 ^d^	0.13 ± 0.01 ^b^	0.18 ± 0.01 ^a^
5 g/L	0.17 ± 0.04 ^d^	0.26 ± 0.05 ^c^	0.13 ± 0.05 ^d^	0.32 ± 0.04 ^b^	0.40 ± 0.01 ^a^
10 g/L	0.30 ± 0.01 ^c^	0.35 ± 0.005 ^c^	0.50 ± 0.05 ^b^	0.70 ± 0.01 ^a^	0.51 ± 0.09 ^b^
20 g/L	0.87 ± 0.01 ^b^	0.82 ± 0.06 ^b^	0.90 ± 0.01 ^a^	0.60 ± 0.03 ^c^	0.80 ± 0.04 ^b^
40 g/L	0.80 ± 0.02 ^a^	0.80 ± 0.08 ^a^	0.63 ± 0.03 ^b^	0.52 ± 0.09 ^c^	0.66 ± 0.02 ^b^

* The OD_600_ value of glucose was 0.03, 0.04, 0.25, 0.58, 0.84, and 0.70 for 1.25 g/L, 2.5 g/L, 5 g/L, 10 g/L, 20 g/L, and 40 g/L, respectively. Values are given as mean ± standard deviation from triplicate determinations. Different letters in the same row indicate significant differences (*p* < 0.05).

## Data Availability

The presented data is contained within the article.

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
