# Peer review of "Formation of Intermediate Amylose Rice Starch–Lipid Complex Assisted by Ultrasonication"

_foods, 2022, doi:10.3390/foods11162430_

Round 1

Reviewer 1 Report

- The study conducted is well planned and the results and discussion is clear with proper citations. They are well presented as well. 

- The materials and methods are well described and research methodology followed is appropriate.

- Overall the manuscript is in acceptable format. 

- I would suggest some minor corrections;

1. Page no. 2 - line 80 - Here write what lipid fractions were added. it is mentioned lipid fractions added.......

2. In tables the letters a,b,c,.... denoting significance should be as superscript

3. In table 1- The light transmittance (%) - alignment

4. In headings and subheadings it would be better to write full name instead of abbrevations such as

    3.2.1 - WAC and OAC

    3.2.3. - LT   etc

Author Response

Reviewer#1

- The study conducted is well planned and the results and discussion is clear with proper citations. They are well presented as well. 

Ans: Thank you very much.

- The materials and methods are well described and research methodology followed is appropriate.

Ans: Thank you very much.

- Overall the manuscript is in acceptable format. 

Ans: Thank you very much.

- I would suggest some minor corrections;

  1. Page no. 2 - line 80 - Here write what lipid fractions were added. it is mentioned lipid fractions added.......

Ans: It is actually “fatty acid”.

  1. In tables the letters a,b,c,.... denoting significance should be as superscript

Ans: Done.

  1. In table 1- The light transmittance (%) – alignment

Ans: Done.

  1. In headings and subheadings it would be better to write full name instead of abbrevations such as

    3.2.1 - WAC and OAC

    3.2.3. - LT   etc

Ans: Done.

Reviewer 2 Report

I have suggested some changes in the attached PDF, which are self-explanatory.

Please consider the following points:

1. The results must be concise and discussed with other updated findings.

2. Conclusion should explain a bit more, and any future study that goes a step forward from the present one should be suggested.

I have no access to check any plagerism software, therefore, please check any similarity, as below few studies are very close to yours:

Zhang, X., Mi, T., Gao, W., Wu, Z., Yuan, C., Cui, B., Dai, Y. and Liu, P., 2022. Ultrasonication effects on physicochemical properties of starch–lipid complex. Food Chemistry388, p.133054.

Bonto, A. P., Tiozon Jr, R. N., Sreenivasulu, N., & Camacho, D. H. (2021). Impact of ultrasonic treatment on rice starch and grain functional properties: A review. Ultrasonics Sonochemistry71, 105383.

Kunyanee, K., Van Ngo, T., Kusumawardani, S., & Lungsakul, N. (2022). Ultrasound-chilling assisted annealing treatment to produce a lower glycemic index of white rice grains with different amylose content. Ultrasonics Sonochemistry, 106055.

Raza, H., Liang, Q., Ameer, K., Ma, H., & Ren, X. (2022). Dual-frequency power ultrasound effects on the complexing index, physicochemical properties, and digestion mechanism of arrowhead starch-lipid complexes. Ultrasonics Sonochemistry84, 105978.

Author Response

Reviewer#2

I have suggested some changes in the attached PDF, which are self-explanatory.

Ans: Every issue brought up in the PDF attachment was carefully addressed. Thank you very much.

Please consider the following points:

  1. The results must be concise and discussed with other updated findings.

Ans: We did our best to make the discussion coherent by using the updated references.

  1. Conclusion should explain a bit more, and any future study that goes a step forward from the present one should be suggested.

Ans: Done.

I have no access to check any plagerism software, therefore, please check any similarity, as below few studies are very close to yours:

Zhang, X., Mi, T., Gao, W., Wu, Z., Yuan, C., Cui, B., Dai, Y. and Liu, P., 2022. Ultrasonication effects on physicochemical properties of starch–lipid complex. Food Chemistry388, p.133054.

Bonto, A. P., Tiozon Jr, R. N., Sreenivasulu, N., & Camacho, D. H. (2021). Impact of ultrasonic treatment on rice starch and grain functional properties: A review. Ultrasonics Sonochemistry71, 105383.

Kunyanee, K., Van Ngo, T., Kusumawardani, S., & Lungsakul, N. (2022). Ultrasound-chilling assisted annealing treatment to produce a lower glycemic index of white rice grains with different amylose content. Ultrasonics Sonochemistry, 106055.

Raza, H., Liang, Q., Ameer, K., Ma, H., & Ren, X. (2022). Dual-frequency power ultrasound effects on the complexing index, physicochemical properties, and digestion mechanism of arrowhead starch-lipid complexes. Ultrasonics Sonochemistry84, 105978.

Ans: Turnitin, which initially checked the similarity index, found no instances of plagiarism in the publications mentioned.

Reviewer 3 Report

This manuscript investigated the formation of intermediate amylose rice starch-lipid complex assisted by ultrasonication. This manuscript is deficient in novelty, and can have a limited contribution to this field. There are a lot of uncertain or erroneous discussions for the effect of operating parameters on the formation of the complexes. This manuscript is suggested to be substantially improved. The issues to be addressed and clarified are listed below.

(1)  Abstract. The abstract should be re-organized to summarize the purpose, methods, results, and conclusion with a concise presentation for this research.

(2)  Introduction. The complexing of starch and lipid has been reported and reviewed in the literature. In this manuscript, it should be further reviewed in more detail with more relevant literatures, including the problem that this research intends to solve.

(3)  Lines 83-85. What type of ultrasound is used in this research? What are the ultrasonic frequency and power for ultra-sonication? This information should be given. What is the temperature during the period of sonication? The detailed procedure and conditions for the ultra-sonication must be given.

(4)  Line 96. What is the definition of complexing index (CI)? The equation for calculating the CI value should be provided.

(5)  Lines 101-103. The detailed procedures and/or equations for the determination of WAC, OAC, swelling index, solubility index, and LT should be given.

(6)  Lines 116-121. The detailed procedures for the determination of bile acid-binding capacity, DPPH, and proliferation rate of bifidobacteria should be given.

(7)  Section 3.1.1. The use of ultra-sonication in complexing of starch and butyric acid is an additional step following the thermal treatment of the control. The authors concluded that the increase of CI values by ultrasonic treatment for 30 min was due to cavitation effect (lines 139-140), and the dramatic decrease in CI value for 60 min of ultrasonic treatment was also due to cavitation, but in excess (lines 146-148). The discussions for the effect of a longer ultrasonic time on the complexing reactions are not convinced, and could lead to a wrong conclusion. What are the definition and meaning of “excess cavitation”? How could the cavitation effect play two roles, increasing and decreasing the complexing reactions, simultaneously? What is the mechanism of the cavitation effect in the complexing reactions of starch and FA?

(8)  Section 3.1.1. The explanation in lines 155-156 cannot be linked to the discussion in 151-152 for a longer time of ultrasonic treatment.

(9)  The full names of the abbreviations used in Fig. 1a-f should be given in the legends, even they have been explained in the text. The tables and figures should be independently readable without referring to the text.

(10)  The qualities of Fig. 1a-f are poor and with unclear conditions in legends. Fig. 1a-f show three different effects, namely ultrasonic amplitude, fatty acid content, and fatty acids. Fig. 1a-f should be divided into three figures to adequately present each effect.

(11)  Section 3.1.2. What is the ultrasonic time in Fig. 1c, d?

(12)  Section 3.1.2. There are many ambiguous and uncertain terms and explanations in Lines 163-167. The discussions for the effect of different ultrasonic amplitudes should be greatly improved. What type of bonding does the “hydrophobic bond” belong to? What are the definition and meaning of “excessive ultrasonic intensity”? What are the functions that different percentages of ultrasonic amplitude can provide? In addition, what are the evidences for the explanation “This finding … by breaking the hydrophobic bond between amylose and FA molecules via the intensified mechanical force…”?

(13)  Section 3.1.2. Lines 171-174. The RS contents can be increased or decreased with different ultrasonic amplitudes. Thus, the explanation “… indicating that … resisted enzymatic digestion.” in lines 173-174 cannot be linked to the explanation “… an increase in RS content with increasing ultrasonic amplitude …followed by a progressive decrease.” in lines 171-172.

(14)  Lines 179-181. Does the ultra-sonication all provide positive effect to facilitate the formation of the complexes, regardless of ultrasonic time and ultrasonic amplitude?

(15)  Lines 190, 193, and 195. Where is the Fig. 3f?

(16)  Lines 195-197. What are the reasons that the SDS will convert to RS at a higher FA concentration (5-7.5%), but not convert to RS at a lower amount of FA (1-2.5%)?

(17)  Lines 242-243. What is the evidence for this explanation “Crystalline regions in RS … amylolytic enzyme attack.”?

(18)  Table 1. What are the conditions for the complexing reactions of each group?

(19)  Section 3.2.1. What are the relationships between SDS and RS in Fig. 1h and WAC and OAC in Table 1 for the comparison of the effect of different FA types?

(20)  From Table 2 and the explanations in lines 338-339 “The formation … viscosity …” and lines 342-345 “The starch-stearic acid complex … resulting in less complex formation.”, these two explanations contradict with each other. What is the linking of the final viscosities of native NK and different types of complexes in Table 2 to their CI values in Fig. 1a and Fig. 1g?

(21)  In Figure 5, only the original SEM images can be acceptable. The SEM images in Fig. 5a-j should be presented by using the original images without any addition of words that cover up any portion of the images.

(22)  Section 3.7. Fig. 5c,e,g,i. The differences in the morphologies of different ultrasonically treated starch-lipid complexes need discussions.

(23)  Section 3.8.1. Lines 468-471. From the comparison of Fig. 1h with Table 3, do the explanations in lines 468-470 contradict the explanation in lines 470-472?

(24)  Section 3.8.1. Lines 474-477. What is the evidence for this conclusion “… ultrasonically treated starch-stearic acid complex … or starch-linoleic acid complexes.”?

Author Response

Reviewer#3

This manuscript investigated the formation of intermediate amylose rice starch-lipid complex assisted by ultrasonication. This manuscript is deficient in novelty, and can have a limited contribution to this field. There are a lot of uncertain or erroneous discussions for the effect of operating parameters on the formation of the complexes. This manuscript is suggested to be substantially improved. The issues to be addressed and clarified are listed below.

  • The abstract should be re-organized to summarize the purpose, methods, results, and conclusion with a concise presentation for this research.

Ans: It was revised.

(2)  Introduction. The complexing of starch and lipid has been reported and reviewed in the literature. In this manuscript, it should be further reviewed in more detail with more relevant literatures, including the problem that this research intends to solve.

Ans: It was revised accordingly and the references were updated.

(3)  Lines 83-85. What type of ultrasound is used in this research? What are the ultrasonic frequency and power for ultra-sonication? This information should be given. What is the temperature during the period of sonication? The detailed procedure and conditions for the ultra-sonication must be given.

Ans: The procedure was detailed. “The paste was cooled down to 60 °C and then ultrasonically treated at a power of 750 W, a single frequency of 20 kHz, for 0-60 min with amplitude of 20% (Model VCX600, Sonics & Materials, Inc., Newtown, CT, USA) and interval pulses of 4 s at room temperature (26-28 °C).”

(4)  Line 96. What is the definition of complexing index (CI)? The equation for calculating the CI value should be provided.

Ans: CI represents the percentage of complexation between starches and lipids [12]. Starch paste (5.0 g) was mixed with 25 mL of distilled water at 50 °C in a 50 mL capped tube. After being vortexed for 2 min, 100 ml of the resulting dispersion was mixed with 15 mL of distilled water, followed by the addition of 2 ml of iodine solution (2.0% KI and 1.3% of I2 in distilled water). Then, the absorbance was read at 690 nm. Pastes made only of starch were used as a reference. To avoid starch retrogradation the tests were conducted within 60 min. CI was calculated as follows:

(5)  Lines 101-103. The detailed procedures and/or equations for the determination of WAC, OAC, swelling index, solubility index, and LT should be given.

Ans: Done.

(6)  Lines 116-121. The detailed procedures for the determination of bile acid-binding capacity, DPPH, and proliferation rate of bifidobacteria should be given.

      Ans: Done.

(7)  Section 3.1.1. The use of ultra-sonication in complexing of starch and butyric acid is an additional step following the thermal treatment of the control. The authors concluded that the increase of CI values by ultrasonic treatment for 30 min was due to cavitation effect (lines 139-140), and the dramatic decrease in CI value for 60 min of ultrasonic treatment was also due to cavitation, but in excess (lines 146-148). The discussions for the effect of a longer ultrasonic time on the complexing reactions are not convinced, and could lead to a wrong conclusion. What are the definition and meaning of “excess cavitation”? How could the cavitation effect play two roles, increasing and decreasing the complexing reactions, simultaneously? What is the mechanism of the cavitation effect in the complexing reactions of starch and FA?

      Ans: Thank you very much. It was changed to “However, the ultrasonically-treated samples for more than 30 min resulted in a reduction in CI values. The report noted that a rise in the release of amylose molecules was seen as being aided by the ultrasonic treatment [5], albeit the precise mechanism of the drop in CI value with longer sonication duration was unknown. Different investigations have shown different ideal ultrasonic times for the released amylose molecules to form complexes with lipid molecules [5, 6].

(8)  Section 3.1.1. The explanation in lines 155-156 cannot be linked to the discussion in 151-152 for a longer time of ultrasonic treatment.

Ans: The explanation in lines 155-156 were deleted.

(9)  The full names of the abbreviations used in Fig. 1a-f should be given in the legends, even they have been explained in the text. The tables and figures should be independently readable without referring to the text.

Ans: Done.

(10)  The qualities of Fig. 1a-f are poor and with unclear conditions in legends. Fig. 1a-f show three different effects, namely ultrasonic amplitude, fatty acid content, and fatty acids. Fig. 1a-f should be divided into three figures to adequately present each effect.

Ans: The qualities and resolutions of Fig. 1a–f have already passed the PACETOOL. It was made the legend more readable. All of Fig. 1a-f can be exhibited together to compare the reaction condition.

(11)  Section 3.1.2. What is the ultrasonic time in Fig. 1c, d?

      Ans: It was stated in the legend that “The reaction was conducted for 30 min to determine the impact of ultrasonic amplitude, fatty acid concentration, and fatty acid chain length.”

(12)  Section 3.1.2. There are many ambiguous and uncertain terms and explanations in Lines 163-167. The discussions for the effect of different ultrasonic amplitudes should be greatly improved. What type of bonding does the “hydrophobic bond” belong to? What are the definition and meaning of “excessive ultrasonic intensity”? What are the functions that different percentages of ultrasonic amplitude can provide? In addition, what are the evidences for the explanation “This finding … by breaking the hydrophobic bond between amylose and FA molecules via the intensified mechanical force…”?

      Ans: Although we made every effort, the phenomenon remained a mystery. The opinion was removed to avoid any confusion. So, the remained content was “An increase in CI values with increasing ultrasonic amplitude up to 20% was observed, followed by a dramatic decrease in CI values (p < 0.05). This finding indicated that excessive ultrasonic intensity had a negative effect on the molecular inclusion of FA into the amylose helix. This may results in the degradation of the amylose helical structure or the final complex structure. The observation was similar to the findings of Liu et al. [17], who revealed that low-power-density ultrasound facilitated in the development of the complex, while increasing ultrasonic amplitude steadily decreased the CI values.”

(13)  Section 3.1.2. Lines 171-174. The RS contents can be increased or decreased with different ultrasonic amplitudes. Thus, the explanation “… indicating that … resisted enzymatic digestion.” in lines 173-174 cannot be linked to the explanation “… an increase in RS content with increasing ultrasonic amplitude …followed by a progressive decrease.” in lines 171-172.

Ans: The statement “… indicating that … resisted enzymatic digestion.” was deleted.

(14)  Lines 179-181. Does the ultra-sonication all provide positive effect to facilitate the formation of the complexes, regardless of ultrasonic time and ultrasonic amplitude?

      Ans: It was changed to “According to the findings, ultrasonic time and amplitude appeared to be crucial variables for facilitating the development of complexes, which restricts amylolytic activity.

(15)  Lines 190, 193, and 195. Where is the Fig. 3f?

Ans: It is actually Fig. 1f.

(16)  Lines 195-197. What are the reasons that the SDS will convert to RS at a higher FA concentration (5-7.5%), but not convert to RS at a lower amount of FA (1-2.5%)?

Ans: The synthesis of the starch-lipid complex with a low amount of free FA (1-2.5%) resulted in the transformation of RDS to SDS with negligible formation of RS (Fig. 1f). This was most likely caused by an inappropriate starch to lipid ratio, which resulted in a low concentration of newly formed RS. The greater extent of RS formation could be obtained by synthesis at high free FA concentrations (5-7.5%), resulting in the conversion of RDS or SDS to RS. However, a decrease in RS content was detected at 10% butyric acid, due to less amylose leaching from the starch granule during the thermal gelatinization. As a result, the establishment of lower amylose-lipid complexes took place.

(17)  Lines 242-243. What is the evidence for this explanation “Crystalline regions in RS … amylolytic enzyme attack.”?

Ans: A reference was added to describe the characteristic of crystalline region. “Granules of starch are typically spherical in shape and feature polycrystalline structures with an amorphous area and crystalline structure. The starch molecules are disorganized in the amorphous zone, but they are precisely arranged in double helices in the crystalline region [16].

(18)  Table 1. What are the conditions for the complexing reactions of each group?

Ans: It was stated in the footnote that “The NK starch was subjected to a 30-min ultrasonic reaction with 7.5% fatty acid at 20% amplitude.

(19)  Section 3.2.1. What are the relationships between SDS and RS in Fig. 1h and WAC and OAC in Table 1 for the comparison of the effect of different FA types?

Ans: The statement was added. “This result was supported by the lower OAC of complexes with higher CI values with improved WAC, like starch-butyric and starch-linoleic complexes (Fig. 1h).

(20)  From Table 2 and the explanations in lines 338-339 “The formation … viscosity …” and lines 342-345 “The starch-stearic acid complex … resulting in less complex formation.”, these two explanations contradict with each other. What is the linking of the final viscosities of native NK and different types of complexes in Table 2 to their CI values in Fig. 1a and Fig. 1g?

Ans: The first explanation in lines 338-339 was deleted and the new statement was “The starch-stearic acid complex with the lowest CI value (Fig. 1g) had the lowest final peak viscosity (p < 0.05), which could be attributed to the steric hindrance power of the C18 chain. Long chain FA, in particular, needs more amylose space to complex, resulting in less complex formation. According to Wang et al. [22], only a part of the C18 sections penetrate the amylose ring system, leaving the rest exposed to the environment. This prevented water from entering the starch granules and significantly inhibited starch gelatinization.”

(21)  In Figure 5, only the original SEM images can be acceptable. The SEM images in Fig. 5a-j should be presented by using the original images without any addition of words that cover up any portion of the images.

Ans: All of the SEM images were the originals. The labels a-j were moved out of the images.

(22)  Section 3.7. Fig. 5c,e,g,i. The differences in the morphologies of different ultrasonically treated starch-lipid complexes need discussions.

Ans: Done. “Depending on the fatty acid employed, the starch-lipid complex had a different morphology. In contrast to smaller butyric acid (Fig. 5c, d), which did not render the starch-lipid complexes with the coated layer in the granules, lauric (Fig. 5e, f), stearic (Fig. 5g, h), and linoleic (Fig. 5i, j) acids did.”

(23)  Section 3.8.1. Lines 468-471. From the comparison of Fig. 1h with Table 3, do the explanations in lines 468-470 contradict the explanation in lines 470-472?

Ans: It was stated that “The increased length of the saturated FA chain led to an increase in bile acid binding capacity (Table 3), which was unrelated to the amount of RS content (Fig. 1h). Linoleic acid inclusion in starch molecules resulted in the lowest bile acid biding capacity among starch-lipid complexes (Table 3), despite having a higher RS content than the starch-stearic acid complex.”

(24)  Section 3.8.1. Lines 474-477. What is the evidence for this conclusion “… ultrasonically treated starch-stearic acid complex … or starch-linoleic acid complexes.”?

Ans: This assumption was removed to avoid the overstatement.

Round 2

Reviewer 3 Report

The authors have provided more detailed information and clarified more clearly in the Materials and Methods of the revised version. But the authors did not clarify and improve some issues (points 1, 13, and 24) in the revised manuscript. In the Results and Discussion, the authors removed and did not strengthen some discussions in the section of 3.1.2. and 3.8.1., leading to the weakening discussion for effects and properties of ultrasound in this study, and this revised version cannot provide the new insights from the revised abstract, discussions and conclusion. In addition, the presentations of standard deviation value (± 0.00) in Table 1 are wrong, and the description of triplicates of data in the section 2.4. must clarify whether the individuals are independent experiments.

Author Response

Reviewer#3

The authors have provided more detailed information and clarified more clearly in the Materials and Methods of the revised version. But the authors did not clarify and improve some issues (points 1, 13, and 24) in the revised manuscript. In the Results and Discussion, the authors removed and did not strengthen some discussions in the section of 3.1.2. and 3.8.1., leading to the weakening discussion for effects and properties of ultrasound in this study, and this revised version cannot provide the new insights from the revised abstract, discussions and conclusion. In addition, the presentations of standard deviation value (± 0.00) in Table 1 are wrong, and the description of triplicates of data in the section 2.4. must clarify whether the individuals are independent experiments.

  • The abstract should be re-organized to summarize the purpose, methods, results, and conclusion with a concise presentation for this research.

Ans: The Abstract has been modified. The background was presented first, followed by the objective, the methods, the key findings, and the conclusion. “Due to the potential reduction in starch availability as well as the production of the distinct physico-chemical characteristics of starch in order to improve health benefits, the formation of starch-lipid complexes has attracted significant attention for improving the quantity of resistant starch (RS) content in starchy based foods. The purpose of this research was to apply ultrasonication to produce intermediate amylose rice (Oryza sativa L.) cv. Noui Khuea (NK) starch-fatty acid (FA) complexes. The effects of ultrasonically synthesized conditions (ultrasonic time, ultrasonic amplitude, FA chain length) on the complexing index (CI) and in vitro digestibility of the starch-FA complex were highlighted. The optimum conditions were 7.5% butyric acid with 20% amplitude for 30 min, as indicated by a high CI and RS contents. The ultrasonically treated starch-butyric complex had the highest RS content of 80.78% with a V-type XRD pattern and an additional FTIR peak at 1,709 cm-1. The increase in water/oil absorption capacity and swelling index were observed in starch-lipid complex. The pasting viscosity and pasting/melting temperatures were lower than those of native starch, despite the fact that it had a distinct morphological structure with a high proportion of flaky and groove forms. The complexes were capable of binding bile acid, scavenging the DPPH radical, and stimulating bifidobacteria proliferation better than native starch, which differed depending on the FA inclusion. Therefore, developing a rice starch-lipid complex can be achieved via ultrasonication.”

(13)  Section 3.1.2. Lines 171-174. The RS contents can be increased or decreased with different ultrasonic amplitudes. Thus, the explanation “… indicating that … resisted enzymatic digestion.” in lines 173-174 cannot be linked to the explanation “… an increase in RS content with increasing ultrasonic amplitude …followed by a progressive decrease.” in lines 171-172.

Ans: This section was revised intensively. Thank you very much.

3.1.2. Effect of Ultrasonic Amplitude

The CI values of ultrasonically-treated NK starch in conclusion with butyric acid by various ultrasonic amplitudes (0, 10, 20, 30, and 40%) are shown in Fig.1c. The CI values of ultrasonically-aided samples were greater than those of the control (0 min, p < 0.05), indicating that ultrasonication enhanced the complexation of starch-butyric acid. An increase in CI values with increasing ultrasonic amplitude up to 20% was observed, followed by a dramatic decrease in CI values (p < 0.05). This finding indicated that excessive ultrasonic intensity had a negative effect on the molecular inclusion of FA into the amylose helix. This may results in the degradation of the amylose helical structure or the final complex structure. The observation was similar to the findings of Liu et al. [25], who revealed that low-power-density ultrasound facilitated in the development of the complex, while increasing ultrasonic amplitude steadily decreased the CI values.

Generally, the starch-lipid complex resisted enzymatic digestion [26]. In ultrasonically treated samples, an increase in RS content with increasing ultrasonic amplitude up to 20% was observed, followed by a progressive decrease (p < 0.05) (Fig. 1d).  It was consistent with the CI values (Fig.1c). It has been reported that the structure of the starch suspension became more homogeneous with the increase of ultrasound amplitude and duration [17]. When ultrasonication was used, swollen starch granules' double helix structure was compelled to open, disrupting their natural semicrystalline structural configuration and revealing their interior lamination structure [9]. Additionally, more linear amylose chains were simultaneously liberated from the starch granules' inner structure and took part in the formation of the FA complex [17]. However, the complex dropped when the ultrasonic amplitude was too high. This decrease was mostly attributed to the violent ultrasound-induced starch chain disintegration [17].

The reduction of RDS and SDS amounts was found in response to changes in RS content (Fig. 1d), suggesting that the digestive starch was converted to an indigestible counterpart. The enzymatic digestibility of ultrasonically-treated samples subjected to amplitudes greater than 20% was increased, which was most likely due to the increased occurrence of free amylose molecules caused by an intense ultrasonic cavitation [6, 25]. These amylose molecules may be susceptible to enzymatic hydrolysis. According to the findings, ultrasonic time and amplitude appeared to be crucial variables for facilitating the development of complexes, which restricts amylolytic activity.

 (24)  Section 3.8.1. Lines 474-477. What is the evidence for this conclusion “… ultrasonically treated starch-stearic acid complex … or starch-linoleic acid complexes.”?

Ans: The new perspective was offered. Also, references were updated.

The bile acid binding capacity indicates cholesterol-lowering activity by increasing fecal bile acid excretion [22]. Table 3 shows the bile acid binding capacities of native NK starch and ultrasonically-treated NK starch with various FA. NK starch-lipid complexes showed significantly higher bile acid binding capacities (12.64-15.21%) than that of native sample (12.02 %). Simsek and El [38] revealed that taro RS had significantly higher bile acid binding capacity (7.6%) than native taro starch (5.2%). As a result, the higher RS content of starch-lipid complexes (Fig. 1h) may be associated with greater bile acid binding capacity when compared to native NK starch. The increased length of the saturated FA chain led to an increase in bile acid binding capacity (Table 3), which was unrelated to the amount of RS content (Fig. 1h). Linoleic acid inclusion in starch molecules resulted in the lowest bile acid biding capacity among starch-lipid complexes (Table 3), despite having a higher RS content than the starch-stearic acid complex. This might be because, in comparison to the more ordered starch-butyric acid or starch-linoleic acid complexes, the ultrasonically treated starch-stearic acid complex has a more irregular structure and more branches, making it simpler to form an emulsion with the bile acid molecules. Typically, bile acids solubilize cholesterol in mixed micelles with phospholipids. Bile acids serve to emulsify fats and aid in their digestion after they are secreted into the intestine [39]. The ability of RS to form inclusion complexes with bile acid was strongly correlated with its double helical structure, suggesting that this is the alternative route through which bile acid may be trapped [40]. According to Abadieet al. [41], the hydrophobic interior of the helical structures generated by the 1,4-linked glucose units of amylose and amylopectin is the most plausible binding site for an amphiphilic ligand with RS. Additionally, the stability of the starch helical complex and the number of binding sites per unit both affect how affine this interaction is [41]. In essence, the presence of lipid molecules causes the starch-lipid complex to have a greater hydrophobic area. Different FA subtypes, though, might provide the complexes varying degrees of hydrophobicity. According to the findings, the bile acid binding capability differed depending on the type of FA. According to reports, RS samples with rather loose interior structures may also be responsible for the most notable improvement in RS's ability to bind bile acids. This could be explained by the hypothesis put forth by Zhou et al. [40] that bile acids may penetrate through the surface fissures of the starch-stearic acid complex, leading to a greater interaction between the RS and bile acids. In comparison to native NK starch, starch-lipid complexes have a greater health-promoting effect because of their potential cholesterol-lowering ability.

The description of triplicates of data in the section 2.4. must clarify whether the individuals are independent experiments

Ans: It was changed to “A completely randomized design was used for experimental design and three independent experiments were performed. All experiments were tested in triplicate and data were recorded as mean ± standard deviation (SD). A probability value of p < 0.05 was significantly deemed. Duncan's multiple range tests were used for mean comparisons and analysis of variance (ANOVA). SPSS for Windows Version 17.0 (SPSS Inc., Chicago, IL, USA) was used.”

In addition, the presentations of standard deviation value (± 0.00) in Table 1 are wrong.

Ans: It was rechecked and the SD were rounding a decimal value to two decimal places.

Round 3

Reviewer 3 Report

The authors have re-organized the abstract, improved the discussions in the sections of 3.1.2. and 3.8.1, clarified the independent experiments in Section 2.4, and re-checked the standard deviation value (± 0.00) in Table 1. However, a minor revision is required as in the following:

(1)  In Table 1, there is still a SD value of (± 0.00) in “Oil absorption capacity”. Please re-check again. It is suggested that the SD value with three decimal places should be used to minimize the rounding error.

Author Response

Reviewer#3

The authors have re-organized the abstract, improved the discussions in the sections of 3.1.2. and 3.8.1, clarified the independent experiments in Section 2.4, and re-checked the standard deviation value (± 0.00) in Table 1. However, a minor revision is required as in the following:

(1)  In Table 1, there is still a SD value of (± 0.00) in “Oil absorption capacity”. Please re-check again. It is suggested that the SD value with three decimal places should be used to minimize the rounding error.

Ans: Thank you very much. It was rechecked and the SD was rounding a decimal value to 3 decimal places, in this case ± 0.001.
